# A comprehensive evaluation of module detection methods for gene expression data

Wouter Saelens [1,2], Robrecht Cannoodt [1,3] & Yvan Saeys[1,2]

A critical step in the analysis of large genome-wide gene expression datasets is the use of module detection methods to group genes into co-expression modules. Because of limitations of classical clustering methods, numerous alternative module detection methods have been proposed, which improve upon clustering by handling co-expression in only a subset of samples, modelling the regulatory network, and/or allowing overlap between modules. In this study we use known regulatory networks to do a comprehensive and robust evaluation of these different methods. Overall, decomposition methods outperform all other strategies, while we do not find a clear advantage of biclustering and network inference-based approaches on large gene expression datasets. Using our evaluation workflow, we also investigate several practical aspects of module detection, such as parameter estimation and the use of alternative similarity measures, and conclude with recommendations for the further development of these methods.

[1] Data Mining and Modelling for Biomedicine, VIB Center for Inflammation Research, 9052 Ghent, Belgium. [2] Department of Applied Mathematics, Computer Science and Statistics, Ghent University, 9000 Ghent, Belgium. [3] Center for Medical Genetics, Ghent University Hospital, 9000 Ghent, Belgium. Correspondence and requests for materials should be addressed to Y.S. (email: yvan.saeys@ugent.be)

Ever since the introduction of genome-wide gene expression profiling technologies, module detection methods have been a cornerstone in the biological interpretation of large gene expression compendia[1–3]. Modules in this context are defined as groups of genes with similar expression profiles, which also tend to be functionally related and co-regulated. Apart from allowing a more global and objective interpretation of gene expression data[4,5], co-expression modules are also frequently used to infer regulatory relationships between transcription factors and putative target genes[6–8]. In addition, modules can improve functional genome annotation through the guilt-by-association principle[9] and allow a better understanding of disease origin[10] and progression[11].

Numerous approaches and algorithms have been proposed for module detection in gene expression data. The most popular approach, clustering, has been used since the first gene expression datasets became available and is still the most widely used to this day[6–8,10]. However, in the context of gene expression, clustering methods suffer from three main drawbacks. First, clustering methods only look at co-expression among all samples. As transcriptional regulation is highly context specific[12], clustering potentially misses local co-expression effects which are present in only a subset of all biological samples. Second, most clustering methods are unable to assign genes to multiple modules. The issue of overlap between modules is especially problematic given the increasing evidence that gene regulation is highly combinatorial and that gene products can participate in multiple pathways[13,14]. A third limitation of clustering methods is that they ignore the regulatory relationships between genes. As the variation in target gene expression can at least be partly explained by variation in transcription factor expression[15], including this information could therefore boost module detection. Several alternative module detection approaches have therefore been developed in order to alleviate these three limitations. Decomposition methods[16] and biclustering[17] try to handle local co-expression and overlap. These methods differ from clustering because they allow that genes within a module do not need to be co-expressed in all biological samples, but that a sample can influence the expression of a module to a certain degree (decomposition methods) or not at all (biclustering methods). Two other alternative methods, direct network inference[15] (direct NI) and iterative NI[18], use the expression data to additionally model the regulatory relationships between the genes.

Given the importance of module detection within the transcriptomics field and the wealth of available methods, it is critical that existing and new approaches are evaluated on objective benchmarks. However, we identified several shortcomings in past evaluation studies[17,19–22], related to the use of multiple evaluation metrics, the correct tuning of parameters, and the biological relevance of synthetic data. In this study we therefore propose a new evaluation pipeline for module detection methods for gene expression data. Central to our approach is that we use known regulatory networks to define sets of known modules, which can be used to directly assess the sensitivity and specificity of the different module detection methods on real data. Using our evaluation strategy we analyze the performance of 42 module detection methods spanning all five main approaches. We also consider several practical aspects of module detection, such as the

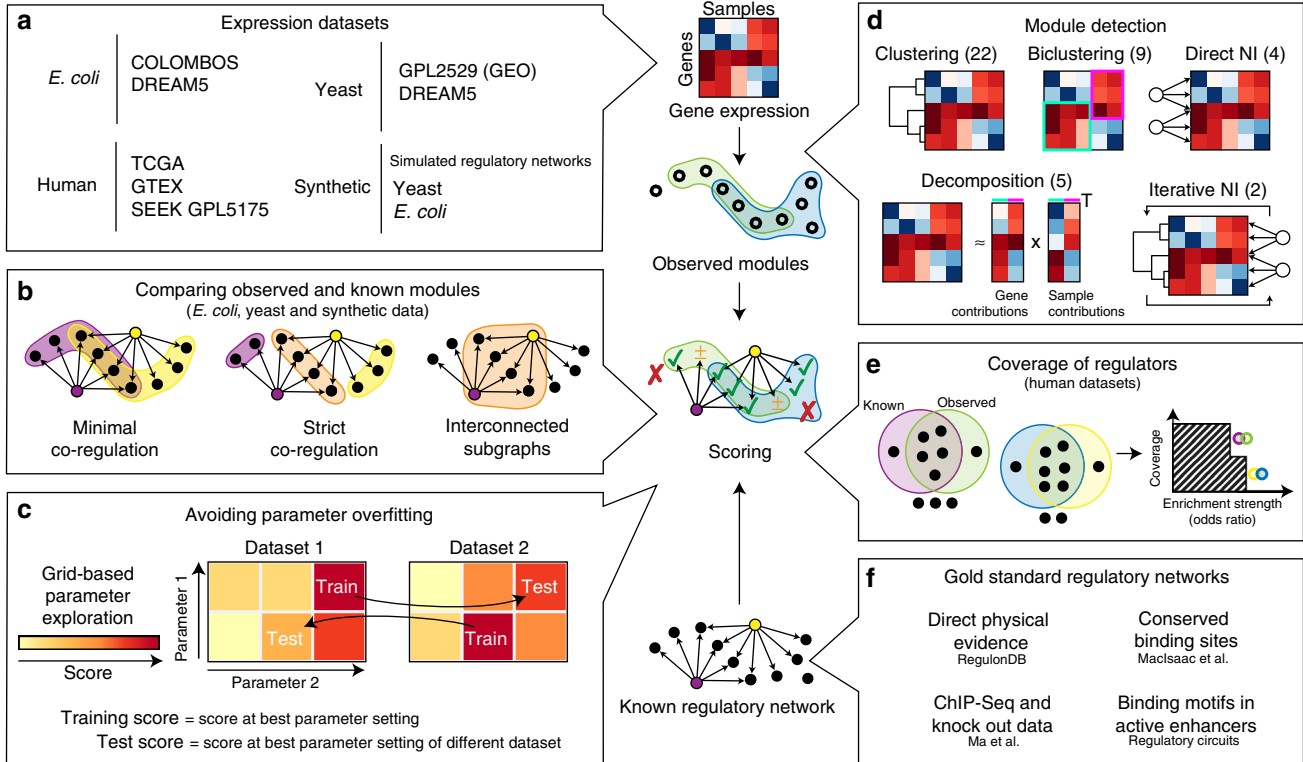

**Fig. 1** Overview of our evaluation methodology. **a** The nine different datasets used in this evaluation. **b** We used three different module definitions to extract known modules from known regulatory networks for the evaluation on *E. coli*, yeast and synthetic data. **c** To avoid parameter overfitting on characteristics of particular datasets, we first optimized the parameters on every dataset using a grid search, and then used the optimal parameters on one dataset (training score) to assess the performance of a method on another dataset (test score). **d** We evaluated a total of 42 methods, which can be classified in 5 categories: clustering, biclustering, direct network inference (NI), decomposition, and iterative NI. **e** For the evaluation on human data, we compared how well the targets of each regulator is enriched in at least one of the modules. **f** We used four different regulatory networks in our evaluation, each generated from different types of data

---

**Table 1 Module detection methods evaluated in this study**

**Clustering: grouping genes based on a global similarity in gene expression profiles**

A  FLAME: fuzzy clustering by selecting cluster supporting objects based on the K-nearest neighbor density estimation

B  K-medoids: iteratively refines the centers (which are individual genes) and the average dissimilarity within the cluster

C  K-medoids (see B) but with automatic module number estimation

D  Fuzzy c-means: similar to k-means (see F), but using fuzzy instead of crisp cluster memberships

E  Self-organizing maps: maps each gene on a node embedded in a two-dimensional graph structure

F  K-means: iteratively refines the mean expression with a cluster and the within-cluster sum of squares

G  MCL: simulates random walks within the co-expression graph by alternative steps of expansion and inflation

H  Spectral clustering: applies K-means in the subspace defined by the eigenvectors of the Pearson's correlation affinity matrix

I  Affinity propagation: clustering by exchange of messages between genes

J  Spectral clustering: applies K-means in the subspace defined by the eigenvectors of the K-nearest-neighbor graph

K  Transitivity clustering: tries to find the transitive co-expression graph in which the total cost of added and removed edges is minimized

L  WGCNA: agglomerative hierarchical clustering (see M), but using the topological overlap measure and a dynamic tree cutting algorithm to implicitly determine the number of modules

M  Agglomerative hierarchical clustering: generates a hierarchical structure by progressively grouping genes and clusters based on their similarity

N  Hybrid hierarchical clustering: combination of agglomerative and divisive hierarchical clustering

O  Divisive hierarchical clustering: generates a hierarchical structure by progressively splitting the genes into clusters

P  Agglomerative hierarchical clustering (see M), but with automatic module number estimation

Q  SOTA: combination of self-organizing maps and divisive hierarchical clustering

R  First finds cluster centers by searching for high-density regions, each gene is then assigned to the cluster of its nearest neighbor of higher density

S  CLICK: uses density estimation to find tight groups of similar genes, after which these are expanded into modules

T  DBSCAN: groups genes within core, non-core and outlier genes based on the number of neighbors

U  Clues: first applies a shrinking procedure which moves each gene towards nearby high-density regions, after which the genes are partitioned into an automatically determined number of clusters using the silhouette width

V  Mean shift: moves each gene towards nearby high density regions until convergence

**Decomposition: extracting the components corresponding to co-expression modules by decomposing the expression matrix in a product of smaller matrices**

A  Independent component analysis: decomposes the expression matrix into a set of independent components using the FastICA algorithm, detects potentially overlapping modules within each source signal using false-discovery rate (FDR) estimation

B  Similar to A, but detects two modules per independent component depending on whether genes have positive or negative weights

C  Similar to A, but detects modules within each source signal using $z$-scores

D  Combination of principal component analysis and independent component analysis, uses FDR estimation to find modules

E  Principal component analysis: decomposes the expression matrix into a set of linearly uncorrelated components, detects potentially overlapping modules within each component using FDR estimation

**Biclustering: simultaneous grouping of genes and samples in biclusters based on similar local behavior in expression**

A  Spectral biclustering: detecting checkerboard patterns within the gene expression matrix

B  ISA: iteratively refines a set of genes and samples based on high or low expression in both the gene and sample dimension

C  QUBIC: finds biclusters in which the genes have similar high or low expression levels in a discretized expression matrix

D  Bi-Force: finds biclusters with over- or under-expression by solving the bicluster editing problem

E  FABIA: builds a multiplicative model of the expression matrix layer by layer. Every layer represents a bicluster

F  Plaid: builds an additive model of the expression matrix layer by layer. Every layer represents a bicluster

G  MSBE: finds additive biclusters starting from randomly sampled reference genes and conditions

H  Cheng & church: minimizes the mean squared residue within every bicluster

I  OPSM: searches for biclusters where the expression changes in the same direction between genes and samples

**Iterative network inference: iterative optimization of an inferred network and a set of clusters**

A  MERLIN: iteratively refines a direct regulatory network and modules within a probabilistic graphical network framework

B  Genomica: starts from an initial hierarchical clustering and iteratively refines this clustering and an inferred module network using a model based on Bayesian regression trees

**Direct network inference: inference of a regulatory network based on gene expression similarity between regulators and target genes**

A  GENIE3: predicts the expression of each target gene based on random forest regression

B  CLR: calculates the likelihood of mutual information estimations based on the network neighborhood

C  Pearson's correlation between regulator and target gene

D  TIGRESS: network inference using a combination of Lasso sparse regression and stability selection

Within each category, methods are ranked according to their average test score (Fig. 2). We refer the reader to Supplementary Note 2 for details regarding the implementation and parameters

---

relative data requirements of the methods, parameter estimation, and the use of alternative similarity measures for clustering. The purpose of this evaluation study is twofold. We first want to provide an overview of the characteristics and performance of current module detection methods to guide the biologist in their choice. Second, we propose a benchmark strategy, which can be used in future studies to compare novel methods with the current state of the art. For this purpose, we provide all gold standards,

expression datasets, and the evaluation procedure to the community.

## Results

**Evaluation workflow**. Our evaluation procedure was structured as follows (Fig. 1). We applied publicly available module detection methods on nine gene expression compendia from *Escherichia coli*,

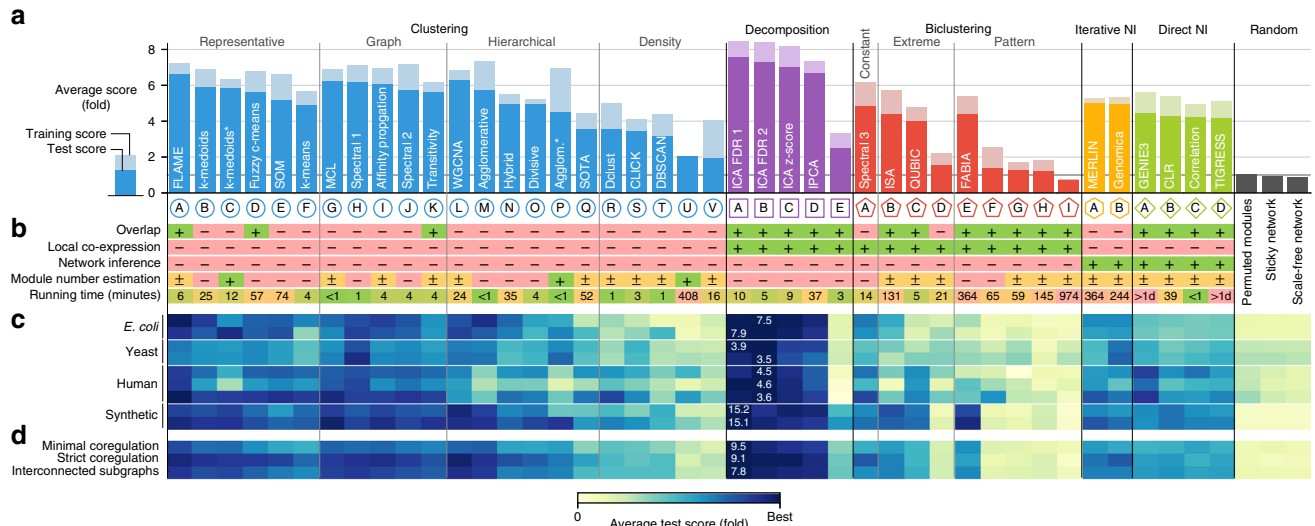

**Fig. 2** Overall performance of 42 module detection methods (Table 1) based on the agreement between observed modules and known modules in gene regulatory networks. The methods can be divided in five categories: clustering, decomposition, biclustering, direct network inference (direct NI) and iterative network inference (iterative NI) methods. Clustering and biclustering methods were further classified in subcategories (see Methods). **a** Average test and training scores across datasets and module definitions. The score represents a fold improvement over permutations of the known modules. *Automatic estimation of number of modules. **b** Different properties of the module detection methods (see Supplementary Note 2). A+ (green background) denotes that a method can handle a certain property listed on the left. We distinguish between explicit (−), implicit (±), and automatic (+) module number estimation. Note that running times strongly depend on the implementation, hardware, dataset dimensions, and parameter settings, and are therefore only indicative. **c** Test scores at each of the four datasets, averaged over module definitions. **d** Test scores on each of the three module definitions, averaged over different datasets

yeast, human, and in silico simulated regulatory networks (Fig. 1a). We scored the different methods by comparing the observed modules with a set of known modules. These known modules were extracted from known regulatory networks using three different module definitions (Fig. 1b), two requiring co-regulation by either one or all known regulators and one looking at strong interconnectedness within the gene regulatory network. To compare a set of observed modules with known modules, we considered several scores described in the literature (Supplementary Note 1) and ultimately chose four scores as follows: recovery, relevance, recall, and precision (Supplementary Fig. 1). Note that classical scores comparing clusterings could not be used because these cannot handle overlap. As all methods generally performed equally or worse than random on human datasets, due to the high number of false positives in the gold standard (Supplementary Note 1), we instead used a scoring system which looks at how well the observed modules cover the targets of regulators in the dataset (Fig. 1e). To avoid certain gold standards and module definitions from disproportionately influencing our final score, we normalized each score using random permutations of the known modules. The final score for a method ultimately represented a fold improvement of a given module detection method over the score obtained from randomly permuted known modules.

Parameter tuning is a necessary but often overlooked challenge with module detection methods. Although good performance generally depends on the correct choice of parameters, this also increases the risk of overfitting on specific characteristics of one dataset, as such parameters will lead to suboptimal results when generalizing the parameter settings to other datasets. To address both problems, we optimized the parameters for every method with a grid search (Supplementary Note 2) and used an approach akin to cross-validation where the optimal parameter settings from one dataset were used to assess the performance of a method on another dataset (Fig. 1c). For every method we give two scores: the training score represents the score at the optimal parameter

settings, whereas the test score denotes the performance when parameters were estimated on an alternative dataset (Fig. 1c).

**Overall performance**. We evaluated a total of 42 module detection algorithms covering all 5 approaches (clustering, decomposition, biclustering, direct NI, and iterative NI) using the described methodology (Table 1 and Supplementary Note 2). Overall, our results indicate that decomposition methods detect the modules which best correspond to the known modular structure within the gene regulatory network (Fig. 2a). The best decomposition methods are all variations of independent component analysis (ICA) with different post-processing methods[16,23]. Surprisingly, neither biclustering nor direct NI, nor iterative NI methods outperform clustering methods, although in theory they should offer several advantages by allowing overlap, modelling transcriptional regulation and/or looking for local co-expression effects (Fig. 2b).

Note that decomposition methods not only perform well when the gold standard modules contains overlap, in the case of minimally co-regulated modules, but also when no overlap is present in the known modules (Fig. 2d). To further investigate this, we calculated separate scores for genes within one or multiple modules. This analysis showed that both clustering and decomposition methods are better at grouping genes that are present in one module, whereas biclustering and direct NI methods are slightly biased toward genes present in more than one module (Supplementary Fig. 2). These results indicate that the higher performance of decomposition methods over clustering is not exclusively caused by their ability to detect overlapping modules, but that also other factors such as local co-expression could have a role.

We further classified clustering algorithms into four categories: graph-based clustering, representative-based clustering, hierarchical clustering, and density-based clustering. We found that graph-based, representative-based, and hierarchical clustering all

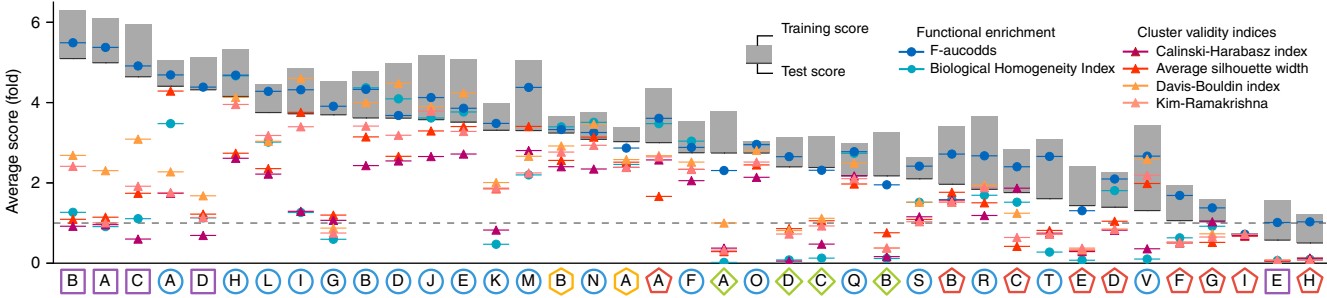

**Fig. 3** Effect of automatic parameter estimation using four different cluster validity indices and two measures based on functional enrichment on the performance of top module detection methods. Shown are changes in test scores after parameter estimation (either using measures based on functional enrichment in blue or cluster validity indices in red–orange), averaged over datasets and module definitions, of the top module detection methods in every category

performed equally well, with the clustering method FLAME (Fuzzy clustering by Local Approximation of Memberships)[24], one of the only clustering methods able to detect overlap, slightly outperforming other clustering methods. Among hierarchical clustering methods, agglomerative methods provide the highest performance compared with the intermediate and low scores of respectively hybrid and divisive methods. Density-based clustering methods on the other hand had much lower performance, which can be partly explained by a higher parameter sensitivity for some density-based methods. Although the overall performance of biclustering methods was low, we also made a similar categorization of these methods based on the type of biclusters they detect. Methods that detect constant or extreme biclusters generally outperformed other methods detecting more complex bicluster patterns. In fact, except for FABIA (Factor Analysis for Bicluster Acquisition), the performance of the latter methods was generally not much better or even worse than random permutations of the known modules.

Although the relative ranking of the main methods is remarkably stable across datasets (Fig. 2c and Supplementary Fig. 3a), individual methods can still perform well in one setting even though their overall performance is poor (Supplementary Fig. 4). Most profoundly is the higher performance of certain biclustering methods, such as ISA (Iterative Signature Algorithm), QUBIC (Qualitative Biclustering), and FABIA, and direct NI methods, primarily GENIE3, on human and/or synthetic data, where these methods can in some cases compete with clustering and decomposition methods. Performance was generally very consistent across different module definitions (Fig. 2d and Supplementary Fig. 3b), despite limited similarity between the sets of known modules (Supplementary Fig. 5). We also found that the overall ranking of the methods remained similar when we used different randomization procedures to normalize our scores (Supplementary Fig. 6). The relative performance of individual methods was more variable when we compared different scores, especially for scores which can handle overlapping modules, although the overall ranking of the different module detection approaches remained stable (Supplementary Figs. 7 and 8). Together, this again highlights the importance of using multiple datasets and scoring metrics for a robust and unbiased evaluation of bioinformatics methods[25,26].

**Parameter tuning**. The need to tune parameters on individual datasets varied greatly among methods, which we quantified by comparing training and test scores. Some methods, such as FLAME, WGCNA (Weighted Gene Co-expression Network Analysis), and MERLIN (Modular regulatory network learning with per gene information) were relatively insensitive to

parameter tuning, despite requiring the optimization of two or more parameters (Fig. 2a). On the other hand, methods such as fuzzy c-means, self-organizing maps, and agglomerative hierarchical clustering performed very differently between test and training parameters. Nonetheless, the overall ranking of the different module detection approaches does not change drastically between training and test scores. The top decomposition methods for instance outperform clustering methods both before and after controlling for parameter overfitting (Fig. 2a and Supplementary Fig. 4).

The most central parameters in module detection (and unsupervised data analysis in general) are those affecting the number of clusters detected within a dataset. We distinguish three different ways a method determines the number of modules (Fig. 2b). Explicit methods, such as k-means and all decomposition methods, require that the number of modules is specified by the user. Implicit methods, such as affinity propagation, adapt the module number on each dataset based on other parameters supplied by the user. Finally, automatic methods determine the number of modules completely automatically, usually by iterating over several parameter settings and selecting the one that optimizes some criterion of cluster quality. Measures for cluster quality can range from the stability of clusters among several resamplings of the dataset[27], the balance between cluster tightness and separateness (as measured by cluster validity indices[28]), or the optimal functional enrichment of the modules[22]. Although the top method within each clustering subcategory estimate the number of modules implicitly (coinciding with a relatively low parameter sensitivity) (Fig. 2b), we found that implicit or automatic estimation of the number of modules is not a prerequisite for a high performance (Supplementary Fig. 9). Indeed, the top decomposition methods all require the number of modules to be specified beforehand. Interestingly, the performance of iterative NI methods depended only little on the initial parameters, possibly because these methods adapt the number of modules depending on the inferred regulatory network.

Apart from those parameters influencing the number of clusters, most module detection methods also have other parameters, frequently affecting the compactness of the modules, the way local co-expression is defined or the minimal number of genes within a module. As all these parameters can have significant effects on the resulting modules (and thus the performance of a method), we assessed how well automatic parameter estimation can improve method performance. Automatic parameter estimation can be seen as an alternative to determining the optimal parameters on one or more training datasets and using these parameters on a test dataset to assess performance. Based on a previous evaluation study[28] we chose four of the most promising cluster validity indices, and found that

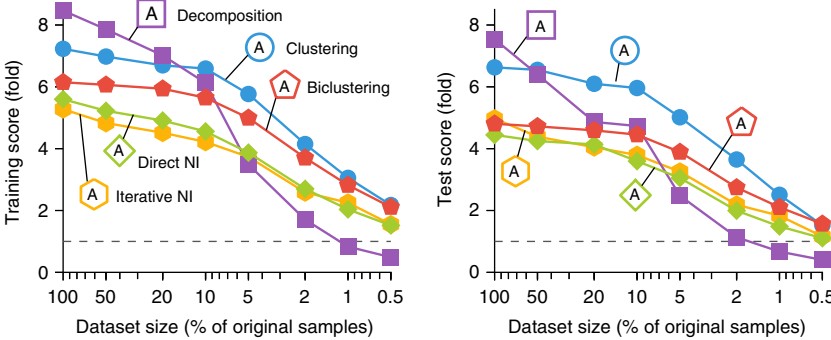

**Fig. 4** Influence of the number of samples on the performance of the top module detection methods. Shown are average training scores (left) and test scores (right) over all datasets and module definitions at different levels of random subsampling (five repeats)

the benefits of cluster validity indices were mostly confined to clustering methods (Fig. 3). Notably, spectral clustering, affinity propagation, and k-medoids frequently increased in test score when their parameters were automatically estimated using the average silhouette width and the Davis–Bouldin index (Supplementary Fig. 10). On the other hand, the performance of FLAME clustering and decomposition methods generally decreased when using cluster validity indices, usually performing even worse than randomly selecting parameters within the parameter grid (Supplementary Fig. 11). It is important to note here that all four cluster validity indices have been developed in the context of clustering methods and were therefore never really designed with overlap and local co-expression in mind, which could explain their low performance with these methods. We also analyzed two alternative measures, which try to estimate the number of clusters based on the optimal enrichment of functional terms or pathways within the modules. We found that a measure that assesses both the coverage of all functional terms as well as the strong individual enrichment of every module (F-aucodds) performed very well, in a large majority of cases performing better than using the optimal parameters of another dataset (Supplementary Fig. 10) and random parameter settings (Supplementary Fig. 11).

Another important parameter for most clustering methods is the distance or similarity measure for comparing gene expression profiles. The most popular metric for gene expression data is undoubtedly the Pearson's correlation coefficient, which measures the extent of the linear dependence between two expression profiles regardless of differences in absolute expression levels. Several authors have criticized this measure[29–31], mainly due to three limitations: (i) it ignores inverse relations between genes (Supplementary Fig. 12a), (ii) it is unable to capture non-linear relationships (Supplementary Fig. 12b), and (iii) it is not robust to outliers and skewed distributions (Supplementary Fig. 12c). Several alternative measures have therefore been proposed, which try to tackle some of these limitations (Supplementary Note 3).

To investigate whether these alternatives are able to improve the module detection, we used 15 such measures as the input for four of the top clustering methods which require having a similarity or distance measure as parameter. Surprisingly, none of the alternative similarity metrics are able to improve performance of any of the four top clustering methods (Supplementary Fig. 12d). When investigating this further, we found several cases where these alternative measures can indeed retrieve known co-regulated genes which were ranked lower than Pearson's correlation, as illustrated with three case examples (Supplementary Fig. 12a–c, e). However, when comparing the top 10% gene pairs between Pearson's correlation and alternative measures, more known co-regulated gene pairs are removed than there are gained (Supplementary Fig. 12f).

**Sensitivity to number of samples and noise**. We next tested the influence of the number of samples within an expression dataset on the relative performance of the top module detection methods within every category. Although, as expected, the performance of every method declined with decreasing dataset size, the magnitude and timing of this decline varied strongly per method. Notably, ICA-based decomposition methods (decomposition methods A and B) seem to be much more sensitive to the number of samples in the dataset compared with other methods (Fig. 4 and Supplementary Fig. 13). On the other hand, the performance of several network inference based methods, such as Genomica (iterative NI method A) and GENIE3 (direct NI method A), remained relatively stable with decreasing number of samples. Together, this indicates that despite its better performance on large datasets, current matrix decomposition methods are unable to meet the performance of clustering methods when a smaller number of biological conditions are being considered.

We also analyzed the noise sensitivity of the different methods by applying different levels of noise on the synthetic datasets. Although we saw that most methods were similarly sensitive to noise compared with their overall performance, we found that some methods, notably WGCNA and fuzzy c-means, were more sensitive (Supplementary Fig. 14).

## Discussion

Unsupervised data analysis has the potential to provide an unbiased and global overview of biological datasets. Compared with other unsupervised clustering tasks in biology (extensively evaluated elsewhere[26]), module detection in gene expression data is unique, because the complexity of the underlying gene regulatory network poses particular challenges, such as local co-expression and overlap. These challenges led to the development of numerous algorithms and tools specifically dedicated to gene expression data; however, so far the comparative performance of these methods was unclear. In this work we therefore introduced a general framework for evaluating module detection methods and used it to provide a first comprehensive evaluation of state-of-the-art module detection methods for gene expression data. Based on this evaluation we analyzed several practical aspects of module detection, such as the choice of methods and parameter estimation, which are summarized in Fig. 5 and will be further discussed here. Moreover, we also provide several guidelines for further development of these methods combined with what in our view has already been accomplished, as summarized in Supplementary Fig. 15.

Module detection in gene expression data can serve a variety of roles and different methods are better suited for particular roles (Fig. 5a,b). Owing to the ease of visualization and interpretation,

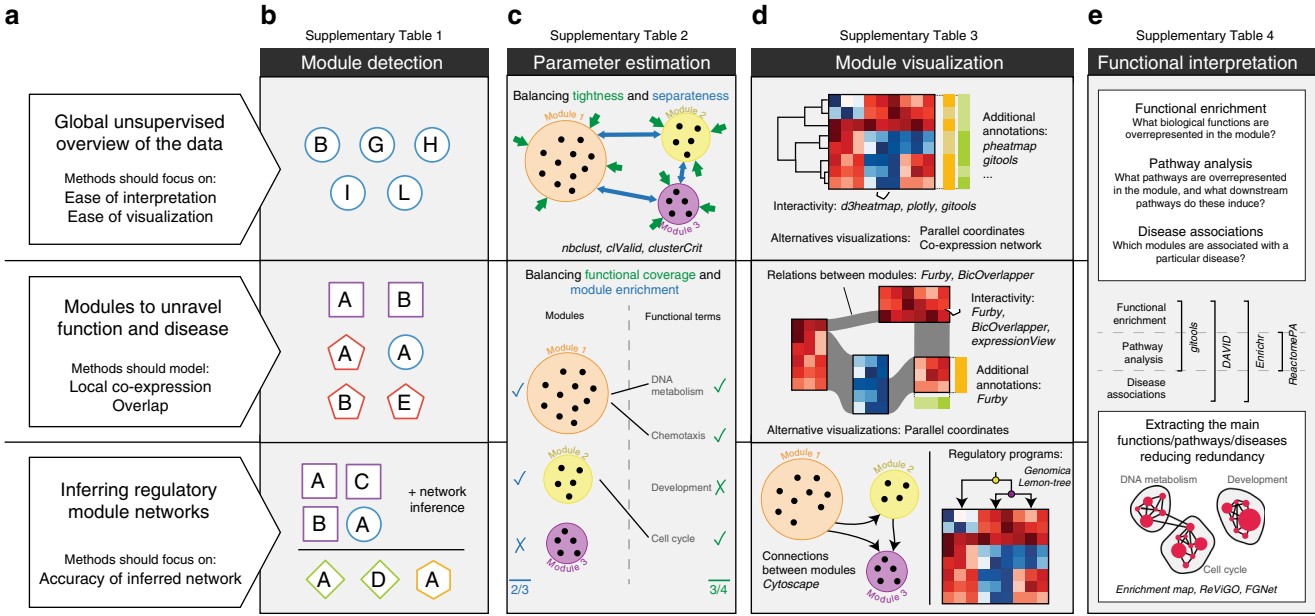

**Fig. 5** Practical guidelines for module detection in gene expression data. Module detection in gene expression data has three main applications (left; panel **a**). For each application, we suggest different module detection methods (**b**), which in turn influences the way parameters are estimated (**c**), how the modules can be visualized (**d**), and how they can be functionally interpreted (**e**)

non-overlapping clustering methods can quickly generate a global overview of the dataset, revealing the main expression and functional effects among the different biological samples in the dataset[2]. Our analysis showed that FLAME, WGCNA, Affinity Propagation, Markov clustering (MCL), and Spectral clustering are particularly suited for such an analysis, outperforming other clustering methods on most datasets. However, because clustering methods do not detect local co-expression effects, they could potentially miss relevant modules or exclude important genes from a module. In use cases where it can be desirable that all modules are discovered in a dataset, e.g., to generate signatures for disease, therapy and prevention[4,11,32], or to find a set of genes responsible for a biological function, methods that detect such local co-expression and/or overlapping modules could therefore provide a substantial advantage. Consistent with this, we found that decomposition methods based on ICA were better at recovering known modules consistently across datasets. Although a handful of studies have already shown the potential of these methods in true biological settings[16,23,32], this was never shown in direct comparison to alternative methods and/or based on objective benchmarks. Finally, a third major application of co-expression modules is in the inference of gene regulatory networks, where modules can be used to improve the network by combining information from several genes[33] but can also improve the ease of interpretation. When we assessed the accuracy of the inferred network by combining a state-of-the-art network inference methods with different module detection methods, we found that ICA-based decomposition methods lead to the highest improvement in accuracy (primarily on yeast and synthetic datasets), closely followed by clustering and graph clustering methods (Supplementary Fig. 16). For most methods, free implementations are available either with a graphical or programming interface, of which we give an overview in Supplementary Table 1.

The choice of method influences subsequent steps of parameter estimation, visualization, and functional interpretation (Fig. 5c–e). For parameter estimation we found that cluster validity indices, the Davis–Bouldin and Kim–Ramakrishna

indices in particular, are sufficient to estimate the parameters for most top clustering methods. However, the performance of these measures on alternative module detection methods was generally worse than randomly selecting parameters. For these methods, biclustering, decomposition, and direct NI in particular, we found that a measure based on functional enrichment provides a better alternative (Fig. 5c). The kind of visualization of the modules also heavily depends on the method. Although the results of a non-overlapping clustering analysis can be readily visualized using heat maps or networks[34], visualizing overlapping modules requires more complex and hierarchical visualizations (which, e.g., indicate the overlap between modules) and is still an active research field[35,36] (Fig. 5d). In both cases, additional annotations can be added to the visualization to improve interpretation of the modules, e.g., to indicate the functional annotation of the genes, and interactivity can be used to accelerate the exploration of the modules. Finally, several tools and databases can be used to functionally interpret the modules, to analyze what biological functions and pathways are enriched within the modules or to find whether the module could be associated with certain diseases. To reduce redundancy in the results of such enrichment analysis, alternative visualization and trimming methods can be used to extract the main biological functions, pathways, or diseases associated with particular modules (Fig. 5e). We give an overview of freely available methods that can be used to interpret co-expression modules in Supplementary Tables 2–5.

Apart from those listed in Fig. 2b, there are also several other characteristics of module detection methods, which are important to consider in practice. Non-exhaustive module detection methods, which include some clustering methods such as FLAME and WGCNA, do not necessarily assign every gene to at least one module. Although this has the advantage that the method itself detects noisy expression profiles, users should be aware that it can also remove a lot of relevant expression profiles if the parameter values are too stringent. Network inference-based methods are unique among the different approaches, because they also generate hypotheses that can explain molecularly why certain genes are grouped in a module. Despite their lower performance

according to our evaluation, they could therefore still be advantageous in certain use cases. Finally, we note that some methods are stochastic, and to assure the robustness of the results users should consider re-running the methods several times in different random states. We list these different properties of a method in Supplementary Note 2, along with a brief discussion about their algorithmic approach, important parameters, and directions to freely available implementations.

As most of the evaluation studies in the past focused only on a limited number of methods, a direct comparison with our evaluation study is difficult[17,19–22]. Furthermore, in these evaluations major conclusions frequently rely on synthetic datasets, and although it can certainly give insights into assumptions made by certain algorithms, it cannot be used to make conclusions about the usefulness of a particular algorithm on real datasets. Still, we acknowledge two noteworthy differences as follows: most studies find that biclustering methods outperform clustering[19,20] or observe substantial performance differences between graph-based, hierarchical, and representative-based clustering methods[21,22]. We relate these differences mainly to issues with parameter estimation, reliance on synthetic datasets and use of a limited number of evaluation metrics. In Supplementary Note 4 we give an overview of past evaluation studies, the methods they evaluated, and the evaluation aspects where we believe these studies are lacking. Similar to a recent study evaluating clustering methods on several biological datasets[26], we found that no single clustering is the best performer on all datasets, although certain methods are certainly better than others at retrieving the known modular structure within the data.

Nonetheless, we acknowledge that our evaluation workflow still has some limitations for particular applications. As we wanted to make sure that most of the modules present in our gold standard were also differentially expressed in the expression data, we used large expression compendia from very different biological conditions. However, this means that when expression differences are very subtle, other methods such as biclustering could perform better. Indeed, some biclustering methods such as FABIA are frequently used in drug discovery[37]. An evaluation focusing on these kind of subproblems is still a possibility for future research.

The detection of overlapping and locally co-expressed modules has been a longstanding challenge in transcriptomics research. Despite great efforts towards the development of these methods, their application on real biological data has been hampered because of several practical challenges. Foremost, the visualization and interpretation of overlapping and locally co-expressed modules is more difficult. Despite some efforts[35,36], current visualization tools, e.g., do not directly show why certain genes are grouped in a module, which can make the module detection methods seem like a black box with unclear biological relevance. Moreover, decomposition and biclustering methods usually have several parameters, which need to be tuned on a dataset and which can affect the biological interpretation. Although we found that external functional information can be used to estimate the parameters of these methods on most datasets, the requirement for such external information can limit their applicability on well-studied organisms. Parameter estimation of biclustering and decomposition methods, which uses only the expression matrix itself, therefore remains an open issue. Finally, our evaluation also indicates that the top performing decomposition methods are much more sensitive to the number of samples in a dataset and are outperformed by clustering methods when the number of samples is limited. We anticipate that improvements on these points (visualization, parameter estimation, and data requirements), will allow these advanced module detection methods to gain more traction in biological research. We list some past accomplishments and points for future research in Supplementary Fig. 15.

## Methods

**Regulatory networks and module definitions**. For *E. coli* datasets, we used a regulatory network from the RegulonDB database version 8 (regulondb.ccg.unam. mx, accessed 03/06/2015), a database integrating both small-scale experimental evidence as well as genome-wide data of transcriptional regulation[38]. We only included interactions with at least one strong evidence type (APPH, BPP, FP, IDA, SM, TA, CHIP-SV, GEA, ROMA, and gSELEX). We did not group the regulatory interactions at operon level, as we found that this has only minimal impact on the overall ranking of the different methods (Supplementary Fig. 17a). We also did not include sigma factor regulations as we found that this would have a negligible effect on performance (Supplementary Fig. 17b). For the yeast datasets we used two regulatory networks. One network was generated from an integration of chromatin immunopurification-on-chip data and conserved binding motifs as described by MacIsaac et al.[39]. Another regulatory network was generated by combining genome-wide transcription factor binding data, knockout expression data, and sequence conservation[40]. We used the most stringent dataset, which required evolutionary conservation in at least two species. For the human datasets we used the 'regulatory circuits' generated by Marbach et al.[41] in which regulators were linked with target genes through a series of steps starting from binding motifs in active enhancers using FANTOM5 project data.

For every gold standard, we obtained sets of known modules based on three different module definitions. We defined minimally co-regulated modules as overlapping groups of genes that shared at least one regulator. Strictly co-regulated modules were defined as groups of genes known to be regulated by exactly the same set of regulators. Strongly interconnected known modules, on the other hand, were defined as groups of genes that are strongly interconnected, and this does therefore not necessarily reflect co-regulation. We used three different graph cluster algorithms (markov clustering, transitivity clustering, and affinity propagation) with in every case three different parameter settings representing different levels of cluster compactness. For the Markov Clustering Algorithm[42] we used inflation parameters 2, 10, and 50. For transitivity clustering[43] we used two different cutoff parameters for the fuzzy membership 0.1 and 0.9. These two parameter settings allowed the modules to overlap (Supplementary Fig. 18). In the third parameter setting for transitivity clustering, we assigned every gene to the module with the highest fuzzy membership value. For affinity propagation[44] we varied the preference value between 0.5, 2, and an automatically estimated value (see Supplementary Note 2). All known modules were then filtered for the genes present in the expression matrix. Finally, we filtered strongly overlapping known modules by merging two modules if they overlapped strongly (Jaccard coefficient > 0.8) and removed small modules by requiring at least five genes. The latter cutoff was defined based on where the average optimal performance of all methods reached a maximum.

To further validate the known modules, we assessed the extent to which the modules are co-expressed in our expression datasets. We found that all three main module definitions generate modules which are both more globally and more locally (according to extreme expression biclustering definition, see Supplementary Note 2) co-expressed compared with permuted modules (Supplementary Fig. 19). Certain module definitions, strict coregulation in particular, and datasets, *E. coli*, and synthetic data generate modules that are better co-expressed within the expression data, which could explain why module detection methods generally also perform better on these datasets and module definitions (Fig. 2c,d). We further confirmed the biological relevance of the known modules by investigating their functional enrichment. We found that on the *E. coli* datasets, 50–70% of all functional terms (both for Gene Ontology (GO)[45] and Kyoto Encyclopedia of Genes and Genomes (KEGG) pathways[46]) were enriched in at least one known module, and that 60–80% of all known modules were enriched in at least one functional term (Supplementary Fig. 20). The coverage of the whole functional space was much less on the yeast data, with about 5–15% GO terms and 10–30% KEGG pathways covered (Supplementary Fig. 20a). On the other hand, a substantial number of all known modules were enriched in at least one functional term, ranging from 30% to 60% on GO terms and 30% to 50% on KEGG pathways (Supplementary Fig. 20b). Compared with known modules, observed modules covered the functional space in most cases a little bit better for the top methods (Supplementary Fig. 21).

**Gene expression data**. We used a total of nine expression datasets for the study, two from *E. coli*, two from *Saccharomyces cerevisae*, three human datasets, and two synthetic datasets. Datasets consisted of hundreds of samples in various genomic and/or environmental perturbational settings.

We obtained a first *E. coli* dataset from the Colombos database (version 2.0, colombos.net)[47]. This dataset is unique among the four because it does not contain raw expression values from one sample but instead contains log ratios between test and reference conditions, which allowed the authors to integrate across different microarray platforms and RNA-sequencing experiments. A second *E. coli* dataset was downloaded from the DREAM5 network inference challenge[15] website (synapse.org/#!Synapse:syn2787209/wiki/70349).

For *S. cerevisiae*, we aggregated an expression compendium by integrating data from 127 experiments (filtered on *S. cerevisae* samples) using the GPL2529 platform from Gene Expression Omnibus (ncbi.nlm.nih.gov/geo). Raw expression data were normalized using Robust Multichip Average as implemented in the

Bioconductor *affy* package. A second yeast dataset was obtained from the DREAM5 website (synapse.org/#!Synapse:syn2787209/wiki/70349).

We obtained the human TCGA datasets from a pan-cancer study of 12 cancer types (synapse.org/#!Synapse:syn1715755)[48]. The human GTEX dataset, which contains expression profiles from different organs from hundreds of donors[49], was downloaded from the GTEX website (gtexportal.org). The SEEK GPL5175 dataset is an aggregation of public datasets using the GPL5175 microarray platform and was retrieved from seek.princeton.edu.

We generated two synthetic datasets starting from the *E. coli* regulondb network and yeast MacIsaac network (both described above) using GeneNetWeaver. This network simulator models the gene regulation using a detailed thermodynamic model and simulates this model using ordinary differential equations[50]. Different experimental conditions were simulated using the 'Multifactorial Perturbations' setting, where transcription rates for a subset of genes are randomly perturbed.

For all expression datasets we filtered out the least variable genes by requiring a minimal standard deviation in expression of 0.5 (for yeast and *E. coli*) and 1 (for human datasets). Heatmaps for every dataset (Supplementary Fig. 21).

Each dataset has its own advantages and disadvantages. Real datasets better fit the real use case and are thus the most biologically relevant, although limited availability of gold standard can make an evaluation on real data challenging. Although our knowledge of the regulatory networks of model micro-organisms, primarily *E. coli*, is already substantial, it is still far from complete[51]. While evaluating on data with more complex regulatory networks such as humans is certainly necessary to ensure the broad relevance of the evaluation, the definition of gold standards on these datasets can be even more problematic because of the broad prevalence of false-positive and false-negative interactions due to a variety of reasons, such as cellular context[12] and non-functional binding[52]. We therefore also included synthetic datasets where the known regulatory network is completely given and thus estimates of both sensitivity and precision of a method can be accurately estimated. Together, we believe these datasets give complementary support to our evaluation strategy and assure its broad relevance.

Similar to a previous evaluation study of biclustering methods[53], our datasets can contain both large differences between samples, as well as small differences, as indicated by the distribution of all log-fold changes between samples (Supplementary Fig. 22).

**Module detection methods**. We chose a total of 42 module detection methods based on (i) a freely available implementation, (ii) performance within previous evaluation studies[17,19–21], and (iii) novelty of the algorithm. See Supplementary Note 2 for a brief overview of every method and Supplementary Table 1 for an overview of the implementations used in this study and alternative implementations. We classified all module detection methods in five major categories. We acknowledge however that the boundaries between the different categories are not always clear, as certain clustering and biclustering methods, e.g., also use a matrix decomposition step within their algorithm. The common theme of clustering methods is that they group genes according to a global similarity in gene expression. Even if clustering methods can detect (after some post-processing) overlapping clusters, this overlap is detected only because a certain gene is still globally similar to both two clusters, and not necessarily because of a local co-expression. Decomposition methods try to approximate the expression matrix using a product of smaller matrices. Two of these matrices contain the individual contributions of respectively genes and samples to a particular module. As samples are allowed to contribute to a particular module only to a certain degree, decomposition methods can detect local co-expression. Related to these methods are biclustering methods, which detect groups of genes, and samples, which show some local co-expression only within the bicluster. In biclustering, samples either contribute to a particular module or not, in contrast to decomposition methods where all samples contribute to a certain extent. Modules detected by biclustering methods can therefore be easier to interpret compared with those of decomposition methods, as the exact origin of the local co-expression is better defined. In some cases, a biclustering method is simply an extension of an existing decomposition method but with an extra requirement that the contribution of a gene and sample to a module is sparse (i.e., contains lots of zeros). Direct NI methods try to generate a simple model of gene regulation, in most cases by using the expression matrix to assign a score to every regulator-gene pair[15]. Although their primary application is to predict novel regulatory relationships between genes, some studies have also used the resulting weighed regulatory network to detect gene modules[54,55]. A list of regulators was generated for *E. coli* by looking for genes annotated by GO with either "transcription, DNA-templated," or "DNA binding," and for yeast and human with "sequence-specific DNA-binding RNA polymerase II transcription factor activity." The same list was also used for iterative NI methods, which start from an initial clustering, and iteratively refine this clustering and an inferred regulatory network.

We further classified clustering methods according to their "induction principle," a classification that does not use the way clusters are represented in the algorithm (the model), but rather looks at the optimization problem underlying the clustering algorithm[56]. Graph-based clustering algorithms make use of graph-like structures, such as K-nearest-neighbor graphs and affinity graphs, and group genes if they are strongly connected within this graph-like structure. Representation-based methods iteratively refine a cluster assignment and representative (such as

the centroid) of the cluster. Hierarchical clustering methods construct a hierarchy of all the genes within the expression matrix. Finally, density-based methods detect modules by looking at contiguous regions of high density. It should also be noted that some clustering methods use elements from multiple categories. FLAME (clustering method A), e.g., uses elements from graph-based, representative-based, and density-based clustering, whereas affinity propagation contains both elements from graph-based and representative-based methods. In cases like this, we ultimately classified an algorithm based on which aspect of the algorithm we believe has the major impact on the final clustering result. Biclustering methods were further classified according to the type of biclusters they detect. The expression within constant biclusters remains relatively stable, whereas the genes within extreme biclusters have a relatively high expression in a subset of conditions compared with other genes. The expression within pattern-based biclusters follow more complex models such as additive models[57], multiplicative models[53], and coherent evolution[58].

Post-processing steps were required for certain methods to get the results in a correct format for comparison with the known modules. All parameters for these post-processing steps were optimized within the grid search approach (as described in Supplementary Note 2 for every method). For fuzzy clustering methods, we obtained crisp but potentially overlapping modules by placing a cutoff on the membership values. For direct NI methods, we first used a cutoff to convert the weighed to an unweighted network, and then detected modules using the same module definitions as previously described. For decomposition methods we explored several post-processing steps in literature (see Supplementary Note 2).

As gene regulatory networks, even in these model organisms, are still very incomplete[51], a small majority of the genes was not included in any known module (Supplementary Fig. 23). Although we did retain these genes in the expression matrix pbefore module detection, we removed these genes in the observed modules before scoring. This was to avoid a strong overestimation of the number of false positives in the observed modules, as most of these genes probably belong to one or more co-regulated modules, which we do not yet know. Finally, similar to the known modules, we filtered the observed modules so that each module contained at least five genes.

Similar to our analysis with known modules, we assessed the extent to which the genes detected by each of the methods are co-expressed in the datasets based on three co-expression metrics inspired by the three types of biclustering metrics (Supplementary Fig. 24). (1) An overall co-expression metric using the average correlation, (2) an extreme expression metric by looking at the top 5% average *z*-scores for every gene in the module, and (3) the root mean-squared deviation within the expression values of each module. For each metric, we compared the distribution of the real modules with permuted modules by calculating the median difference using the wilcox.test function in R. We found that every module detection method found modules, which were more strongly co-expressed than permuted modules. Compared with the co-expression of known modules, the module detection methods also produced modules that are more strongly co-expressed. Specifically for biclustering methods, we also investigated the co-expression only in those samples within each bicluster. Here we found that, except for some pattern-based biclustering methods, most biclustering methods detected the type of modules, which they are designed to detect (Supplementary Fig. 24).

**Parameter tuning**. Parameter tuning is a necessary but often overlooked challenge with module detection methods. All too often, evaluation studies use default parameters which were optimized for some specific test cases by the authors. This does not correspond well with the true biological setting, where some parameter optimization is almost always necessary to make sense of the data. Therefore, to make sure an evaluation is as unbiased as possible, some parameter optimization is always required. However, one should be careful of overfitting parameters on specific characteristics of one dataset, as such parameters will lead to suboptimal results when generalizing the parameter settings to other datasets. This could again introduce a bias in the analysis, where methods with a lot of parameters would better adapt on particular datasets, but would not generalize well on other datasets. In this study we tried to address both problems as follows. We first used a grid search to explore the parameter space of every method and determine their most optimal parameters given a certain dataset and module definition, which resulted in a training scores. Next, in a process akin to cross-validation, we used the optimal parameters of one training dataset from another organism to score the performance on another test dataset, which resulted in a test scores for every training dataset. As we saw that optimal parameters were in most cases very different between synthetic and real datasets, we only used real datasets to train parameters for other real datasets and synthetic datasets for other synthetic datasets. We refer to Supplementary Note 2 for an overview of what parameters were varied for every method.

**Evaluation metrics**. We used four different scores to compare a set of known modules with a set of observed modules and, after normalization, combined them in one overall score. Note that classical scores comparing clusterings, such as the Rand index, the F1, or the normalized mutual information, could not be used as these scores are unable to handle overlap and/or overlap[59] (Supplementary Note 1).

The recall and specificity within the recently proposed CICE-BCubed scoring system measure whether the number of modules containing a certain pair of genes is comparable between the known and observed modules[60]. It is based on the Extended BCubed[59], but reaches the perfect score of 1 only when both known and observed overlapping clusterings are equal. If $G$ represents all genes, $M$ a set of known modules, $M'$ a set of observed modules, $M(g)$ the modules that contain $g$, and $E(g, M)$ the set of genes that are together with $g$ in at least one module of $M$ (including $g$ itself), the precision is defined as:

$$\text{Precision} = \frac{1}{|G|} \sum_{g \in G} \frac{1}{|E(g, M')|}$$

$$\sum_{g' \in E(g, M')} \frac{\min(|M'(g) \cap M'(g')|, |M(g) \cap M(g')|) \cdot \Phi(g, g')}{|M'(g) \cap M'(g')|}$$

where

$$\Phi(g, g') = \frac{1}{|M'(g, g')|} \sum_{m' \in M'(g, g')} \max_{m \in M(g, g')} \text{Jaccard}(m', m)$$

Recall is calculated in the same way but with $M'$ and $M$ switched. The recovery and relevance scores, which have been previously used in evaluation studies of biclustering methods, assess whether each observed module can be matched with a known module and vice versa[17,19]. Relevance is defined as

$$\text{Relevance} = \frac{1}{|M'|} \sum_{m' \in M'} \max_{m \in M} \text{Jaccard}(m', m)$$

Recovery is calculated in the same way but with $M'$ and $M$ switched.

Before combining scores across different datasets and module definitions, we first normalized every score by dividing it by an average score of 500 permuted versions of the known modules (Supplementary Fig. 25). The goal of this step was to prevent easier module structures (small modules, low number of modules, and no overlap) of certain module definitions and datasets from disproportionally influencing the final score. Permuted modules were generated by randomly mapping the genes of a dataset to a random permutation of the genes and replacing every occurrence of a gene in a known module with its mapped version. Based on this random model, module structure (size, number, and overlap) remained the same, while only the assignment changed. We also tested two alternative random models. The STICKY random model has been previously described[61]. We adapted this model for directed networks by calculating the stickiness index separately for incoming and outgoing edges. For the scale-free network[62], we used the networkx Python package with default parameters.

We finally calculated the harmonic mean between the normalized versions of all four scores to obtain a final score representing the performance of a particular method on a given dataset and module definition.

For human data we used an alternative score that assesses the extent to which the targets of every regulator are enriched with at least one module of the dataset. As described earlier, we used the clustered version of the regulatory circuits dataset[41], which contains weights for every regulator and target gene combination across 32 tissue and cell-type contexts. For every combination of target genes and observed module we calculated a $p$-value of enrichment using a right-tailed Fisher's exact test (corrected for multiple testing using the Holm–Šídák procedure[63]) and the strength of this enrichment using the odds ratio. Although we calculated these values within every cell type and tissue context separately, we retained for every regulator its minimal $p$-value and the corresponding odds ratio across the different contexts, as we do not know the exact cell-type and tissue context in which the genes of the observed modules are co-expressed. We then extracted for every regulator its maximal odds ratio across the observed modules where the targets of the regulators were enriched (corrected $p$-value < 0.1). The aucodds score was then calculated by measuring the area under the curve formed by the percentage of regulators with an odds ratio equal or larger than a particular cutoff and the $\log_{10}$ odds ratios within the interval 1 and 1000-fold enrichment. To work in a cutoff-independent manner we averaged the aucodds scores over a range of weight cutoffs. Performance generally decreased with more stringent cutoffs (Supplementary Fig. 26a,b) although some biclustering methods and direct NI methods remained more stable across a wide range of cutoff values (Supplementary Fig. 26c,d). This score was normalized in a similar way as previously described, where the initial known modules were defined using the minimal co-regulation module definition and subsequently randomly permuted by mapping the genes within the modules to a random permutation.

We reweighted the scores between datasets and module definitions using a weighted mean so that module definitions (minimal co-regulation, strict co-regulation, and interconnected subgraphs) and each organism (E. coli, yeast, human, and synthetic) had equal influence on the final score.

**Influence of overlap**. We split the genes of every datasets based on whether they belonged to only one or multiple modules using the minimal co-regulation module definition. If $G^*$ denotes such a subset of genes in the expression matrix, we calculated a precision* score specifically for this subset using:

$$\text{Precision}^* = \frac{1}{|G^*|} \sum_{g \in G^*} \frac{1}{|E(g, M')|}$$

$$\sum_{g' \in E(g, M')} \frac{\min(|M'(g) \cap M'(g')|, |M(g) \cap M(g')|) \cdot \Phi(g, g')}{|M'(g) \cap M'(g')|}$$

A Recall* score was calculated similarly but with $M'$ and $M$ switched. A final score for a particular set of genes was obtained by taking the harmonic mean between the normalized versions of the Recall* and Precision*.

**Automatic parameter estimation**. The four cluster validity indices evaluated in this study all performed well in a recent evaluation study and are defined there[28]. Most indices try to optimize the balance between tightness (the expression variability within a module) and separation (the expression differences between modules). For metrics requiring a distance matrix, we subtracted the absolute Pearson's correlation from one.

We also investigated two metrics to assess the functional coherence of the modules according to the GO database[45] and the KEGG pathways database[46]. We filtered redundant gene sets by, starting from the largest gene set, removing gene sets if they overlap too much with larger non-removed gene sets (Jaccard index > 0.7). The biological homogeneity index measures the proportion of gene pairs within a module which are also matched within a functional class[22]. For the F-aucodds score we calculated an aucodds score as described earlier in both the dimension of the gene sets (denoting how well all functional sets are covered by the observed modules) and the dimension of the observed modules (denoting how well the modules are enriched in at least one function gene set), and combined both scores by calculating its harmonic mean.

As automatic parameter estimation performed very poorly on non-exhaustive module detection methods (which include some clustering methods, see Supplementary Note 2), we assigned every unassigned gene to the module with which the average correlation was the highest prior to calculating the cluster validity indices.

**Similarity measures**. For clustering methods requiring a similarity matrix as input, we used the Pearson's correlation in our evaluation. For methods requiring a dissimilarity matrix, we subtracted the Pearson's correlation values from two. For the comparison of different similarity measures, we selected four top clustering methods that require a similarity measure as input. We compared a total of 10 alternative measures that are briefly described in Supplementary Note 3 along with directions to implementations. We did not evaluate the distance correlation, percentage bend correlation, Hoeffding's D, and maximal information coefficient[64], because they required excessive amounts of computational time and/or memory, which would be impractical for module detection in general use cases. To convert a similarity matrix to a dissimilarity matrix or vice versa, we subtracted the values from the maximal value between all gene pairs on a given dataset. To determine the influence of an alternative similarity measure on the performance of clustering methods, we re-ran all parameter optimization steps for every alternative measure and again calculated test scores as described earlier.

**Code availability**. Code to evaluate module detection methods and further expand the evaluation are available as Jupyter Notebooks[65,66] (jupyter.org) at www.github.com/saeyslab/moduledetection-evaluation.

**Data availability**. Data is available in a Zenodo repository[67] (doi: 10.5281/zenodo.1157938).

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

## Acknowledgements

This study was supported by Ghent University (W.S., B.N.L., and Y.S.) and Fonds Wetenschappelijk Onderzoek (W.S. and R.C.). Y.S. is an ISAC Marylou Ingram Scholar.

## Author contributions

W.S. and Y.S. designed the study. W.S. performed the study. Y.S. supervised the work. W.S., R.C., and Y.S. wrote the paper.

## Additional information

**Competing interests:** The authors declare no competing interests.

