## [Peer Review File(PDF 441 kb) · Nature Communications]

Reviewers' comments:

Reviewer #2 (Remarks to the Author):

I had previously reviewed this paper and found this to be a very useful piece of work. My comments have been addressed in this new version. There are a few minor comments that need to be addressed. I also read through the comments from the other reviewers and I offer my opinion on ways in which the authors can improve their response to these comments:

Specific minor comments:

1. The legend for supp fig 4c in supplementary note 1 is missing and needs to be added.
2. The legend of supp fig 21 should mention what clustering algorithm is used on the output of the direct NI methods prior to enrichment analysis.
3. The authors use f-aucodds score to assess the functional enrichment of their modules. This type of analysis usually uses a hypergeometric test based p-value. Can the authors report the p-values or provide a justification why this is not possible.
4. While clustering and network inference methods are clearly distinct from biclustering and decomposition based methods, I think the distinction between biclustering and decomposition-based methods is not clear. Although the authors explicitly acknowledge this in the text ("Module detection methods"), this could be certainly expanded to make clear the distinction between decomposition and biclustering. It seems what the authors are calling decomposition methods are those that are doing dimensionality reduction only in the sample space, but biclustering methods are doing dimensionality reduction in both the gene and sample space. Is this correct? Could the authors clarify this distinction more clearly in the text and perhaps in figure 1b ?

Addressing Reviewer 1's comment:

Reviewer 1's initial main concern was whether what the authors define as gold standard modules is apparent in the expression data that the authors are applying different methods on.

To address this concern the authors measured the pairwise correlations between genes in the same module and showed that the genes are more co-expressed than a random set of modules. They call this "global" co-expression (Supp fig 19a). Because bi-clustering methods are aiming to identify local co-expression, they also measure the absolute value of z-scores and report the 95 percentile z-score for each module. The distribution of these z-scores is much higher compared to random modules from which one can conclude that the ground truth modules are more locally co-expressed (and therefore biclustering methods should perform well here).

Reviewer 1's concern now is that the authors use correlation for assessing global modules and a different statistic (extreme expression metric defined by the 95% percentile absolute value of z-score)

for the local modules. The authors' justification for using correlation is that most biclustering methods do not use correlation, and, in fact, methods such as ISA and QUBIC use extreme expression. In my opinion, the author's response can be improved by providing a comparison based on expression coherence in a module. Perhaps the advantage of bi-clustering methods would be more obvious if the methods were compared based on the extent to which the modules identified by each method is also co-expressed. Furthermore, I think the z-score analysis in fig 19b is confusing and does not discriminate between pairs that are highly correlated or anti-correlated. This is an important distinction to make.

Reviewer 1's concern can be addressed by doing the following: (a) include a measure of statistical coherence, in addition to the topological overlap of modules, which the authors are currently using. The authors can use correlation or extreme expression but instead of the 5% of samples, the authors use the biclusters selected by each method. (b) compare the distribution of this statistic in the inferred modules to those in the gold standard modules. If the methods are finding good modules then the distribution of this statistic of the inferred and gold standard modules would be similar or the inferred modules should be at least as co-expressed/extremely expressed as the gold standard modules. (c) Rank methods based on the measure of statistical coherence in the modules and also show where the ground truth modules rank using this measure.

I suggest the authors incorporate this into supp fig 19 a and provide a summary statistic (e.g. the p-value from a random test measuring the difference between the foreground and background distributions for each method). This need not be integrated with Fig 1, where the point is to see if modules defined based on network structure can be recovered from data. What this will however demonstrate is the extent to which different methods capture spurious versus true statistical dependencies (note that some true dependencies may not be reflected in the module identified from the network).

Addressing Reviewer 4's comment:

Here the main concern is again relevant to the bi-clustering methods not performing well. The reviewer feels that the specific datasets that have been used might not be well-suited for the bi-clustering methods and likely that is why they do not perform well.

The authors response is sufficient but could be improved by clarifying the differences between bi-clustering methods that use matrix factorization (e.g. FABIA, spectral clustering) and other methods. Specifically, it appears that some of the bi-clustering methods (e.g. spectral bi-clustering and the FABIA method) do quite well. For the other methods, I think the discrepancy is between what the authors are calling "gold standard modules" that are derived from the network structure, and what the bi-clustering methods are trying to learn from the data. When there is a mismatch between these two, the methods obviously do not look good. This is not an issue with this paper and is indeed expected. This is why the independent measure of looking at functional enrichment offers a different and independent way of assessing the modules as it is different from any of the objectives each of the methods are analyzing. This needs to be more clearly explained.

The authors are also correct that the Hochreiter et al paper only compared different biclustering methods and only showed the superiority of their FABIA method over other biclustering methods. Hence, the current manuscript is much broader in scope of their analysis and claims.

Therefore I think the disagreement between the reviews can be reconciled in the following way:

a) The authors provide a ranking of the methods based on a measure of expression coherence within a module (this is what reviewer 1 is asking and I agree with them). See my comment about Reviewer

1's concern.

b) The authors update the introduction/discussion acknowledging that the poor performance for a method can also be explained by specific statistical properties of the data that each method is suited for.

c) I agree with reviewer 1's original comment that an expression matrix clustered by rows and columns would still reveal a useful picture. I suggest the authors try to do this using two or three ways: one simple way is to cluster the rows and columns independently and order them. The other two ways can be based on SVD or one of the best performing decomposition based methods.

Please note that while Reviewer #3 doesn't have remarks to the authors, in his/her remarks to editors, he/she says his/her comments have been sufficiently addressed.

Reviewer #4 (Remarks to the Author):

I thank the authors for their clarification. I have to admit that I was wrong that no comparisons with standard clustering are performed in Hochreiter's FABIA paper. However, the authors' reply is not adequately addressing the point that I made. I was neither referring to small data sets in terms of the number of samples, nor was I referring to data sets in which the genes' fold changes are small. I was referring to data sets in which only a small number of pathways/gene modules is differentially expressed, e.g. samples from the same cell lines, but in slightly different cell states or treated with different, but similar, drugs. This does not seem to be the case in the benchmark data sets considered in this study. Consider, for instance, the study of Verbist et al. (Drug Discovery Today 20, 2015) which also highlights that biclustering has become a standard part of the data analysis pipelines of pharma companies. I fear that the given paper will be misunderstood by the community in the way that biclustering is not a worthwhile tool at all, which is definitely not the case and has been demonstrated in many previous studies.

Reviewers' comments:

Reviewer #2 (Remarks to the Author):

I had previously reviewed this paper and found this to be a very useful piece of work. My comments have been addressed in this new version. There are a few minor comments that need to be addressed. I also read through the comments from the other reviewers and I offer my opinion on ways in which the authors can improve their response to these comments:

Specific minor comments:

1. The legend for supp fig 4c in supplementary note 1 is missing and needs to be added.

This was added, thank you.

2. The legend of supp fig 21 should mention what clustering algorithm is used on the output of the direct NI methods prior to enrichment analysis.

The clustering algorithm is a parameter (apcluster, transitivity clustering, or mcl), and this the choice of clustering method depends on the one that gave the most optimal performance given the training data. In most cases this was markov clustering.

3. The authors use f-aucodds score to assess the functional enrichment of their modules. This type of analysis usually uses a hypergeometric test based p-value. Can the authors report the p-values or provide a justification why this is not possible.

To calculate the f-aucodds, we combine both the p-values (corrected for multiple testing) and the strength of enrichment (given by the odds-ratio). Although related, one tells us the statistical significance of the enrichment, while the other tells us how strong this enrichment is. We decided to combine these two measures using a simple approach, by first only taking into account those odds-scores which are statistically significant (FDR-corrected p-value lower than 0.1), and then calculating the strength of upregulation for every gene set using an area under the curve. Although it is certainly possible to report the individual p-values, we opted for this approach as it assures statistical significance (if no gene set is significant, the aucodds score will also be zero), but also assesses how strong the enrichment is.

4. While clustering and network inference methods are clearly distinct from biclustering and decomposition based methods, I think the distinction between biclustering and decomposition-based methods is not clear. Although the authors explicitly acknowledge this in the text ("Module detection methods"), this could be certainly expanded to make clear the distinction between decomposition and biclustering. It seems what the authors are calling decomposition methods are those that are doing dimensionality reduction only in the sample space, but biclustering methods are doing dimensionality reduction in both the gene and sample space. Is this correct? Could the authors clarify this distinction more clearly in the text and perhaps in figure 1b ?

Thank you for this remark. As this distinction is crucial to understand the rest of the paper, we added a brief sentence explaining the main difference between these two methods in the introduction section:

"Decomposition methods and biclustering try to handle local co-expression and overlap. These methods differ from clustering because they allow that genes within a

module do not need to be co-expressed in all biological samples, but that a sample can influence the expression of a module to a certain degree (decomposition methods) or not at all (biclustering methods)."

We also updated Figure 1b to make the decomposition of an expression matrix clearer. Finally, we also added an extra sentence to the "module detection methods" section, to explain how biclustering methods can be seen as an extension of decomposition methods:

"In some cases, a biclustering method is simply an extension of an existing decomposition method but with an extra requirement that the contribution of a gene and sample to a module is sparse (i.e. contains lots of zeros)."

Addressing Reviewer 1's comment:

Reviewer 1's initial main concern was whether what the authors define as gold standard modules is apparent in the expression data that the authors are applying different methods on.

To address this concern the authors measured the pairwise correlations between genes in the same module and showed that the genes are more co-expressed than a random set of modules. They call this "global" co-expression (Supp fig 19a). Because bi-clustering methods are aiming to identify local co-expression, they also measure the absolute value of z-scores and report the 95 percentile z-score for each module. The distribution of these z-scores is much higher compared to random modules from which one can conclude that the ground truth modules are more locally co-expressed (and therefore biclustering methods should perform well here).

Reviewer 1's concern now is that the authors use correlation for assessing global modules and a different statistic (extreme expression metric defined by the 95% percentile absolute value of z-score) for the local modules. The authors' justification for using correlation is that most biclustering methods do not use correlation, and, in fact, methods such as ISA and QUBIC use extreme expression. In my opinion, the author's response can be improved by providing a comparison based on expression coherence in a module. Perhaps the advantage of bi-clustering methods would be more obvious if the methods were compared based on the extent to which the modules identified by each method is also co-expressed. Furthermore, I think the z-score analysis in fig 19b is confusing and does not discriminate between pairs that are highly correlated or anti-correlated. This is an important distinction to make.

Reviewer 1's concern can be addressed by doing the following: (a) include a measure of statistical coherence, in addition to the topological overlap of modules, which the authors are currently using. The authors can use correlation or extreme expression but instead of the 5% of samples, the authors use the biclusters selected by each method. (b) compare the distribution of this statistic in the inferred modules to those in the gold standard modules. If the methods are finding good modules then the distribution of this statistic of the inferred and gold standard modules would be similar or the inferred modules should be at least as co-expressed/extremely expressed as the gold standard modules. (c) Rank methods based on the measure of statistical coherence in the modules and also show where the ground truth modules rank using this measure.

I suggest the authors incorporate this into supp fig 19 a and provide a summary statistic (e.g. the p-value from a random test measuring the difference between the foreground and background distributions for each method). This need not be integrated with Fig 1, where the point is to see if modules defined based

on network structure can be recovered from data. What this will however demonstrate is the extent to which different methods capture spurious versus true statistical dependencies (note that some true dependencies may not be reflected in the module identified from the network).

We kindly thank the reviewer for these suggestions. We agree that an analysis investigating how strongly the modules found by each method are co-expressed is warranted. Although we have to warn about overinterpreting these results, as these scores have a similar issue as scores based on functional enrichment because they do not directly look at the “sensitivity” of a method. A method which only discovers a small number of highly co-expressed modules will have a high score, but does not necessarily provide a complete representation of the data.

We included this analysis as Supplementary Figure 24 and provide a discussion in the online methods:

“Similar as to our analysis with known modules, we assessed the extent to which the genes detected by each of the methods are co-expressed in each of the datasets based on three co-expression metrics inspired by the three types of biclustering metrics (Supplementary Figure 24). (1) An overall co-expression metric using the average correlation, (2) an extreme expression metric by looking at the top 5% average z-scores for every gene in the module and (3) the root mean squared deviation within the expression values of each module. For each metric, we compared the distribution of the real modules with permuted modules by calculating the median difference using the wilcox.test function in R. We found that every module detection method found modules which were more strongly co-expressed than permuted modules. Compared to the co-expression of known modules, the module detection methods also produced modules which are more strongly co-expressed. Specifically for biclustering methods, we also investigated the co-expression only in those samples within each bicluster. Here we found that, except for some pattern-based biclustering methods, most biclustering methods detected the type of modules which they are designed to detect (Supplementary Figure 24).”

Addressing Reviewer 4's comment:

Here the main concern is again relevant to the bi-clustering methods not performing well. The reviewer feels that the specific datasets that have been used might not be well-suited for the bi-clustering methods and likely that is why they do not perform well.

The authors response is sufficient but could be improved by clarifying the differences between bi-clustering methods that use matrix factorization (e.g. FABIA, spectral clustering) and other methods. Specifically, it appears that some of the bi-clustering methods (e.g. spectral bi-clustering and the FABIA method) do quite well. For the other methods, I think the discrepancy is between what the authors are calling "gold standard modules" that are derived from the network structure, and what the bi-clustering methods are trying to learn from the data. When there is a mismatch between these two, the methods obviously do not look good. This is not an issue with this paper and is indeed expected.

This is why the independent measure of looking at functional enrichment offers a different and independent way of assessing the modules as it is different from any of the objectives each of the methods are analyzing. This needs to be more clearly explained.

The authors are also correct that the Hochreiter et al paper only compared different biclustering methods and only showed the superiority of their FABIA method over other biclustering methods. Hence, the current manuscript is much broader in scope of their analysis and claims.

Therefore I think the disagreement between the reviews can be reconciled in the following way:

a) The authors provide a ranking of the methods based on a measure of expression coherence within a module (this is what reviewer 1 is asking and I agree with them). See my comment about Reviewer 1's concern.

Thank you, this was addressed above.

b) The authors update the introduction/discussion acknowledging that the poor performance for a method can also be explained by specific statistical properties of the data that each method is suited for.

Thank you for this suggestions. We dedicated a paragraph in the discussion for discussing this:

“Nonetheless, we acknowledge that our evaluation workflow still has some limitations for particular applications. Because we wanted to make sure that most of the modules present in our gold standard were also differentially expressed in the expression data, we used large expression compendia from very different biological conditions. However, this means that when expression differences are very subtle, other methods such as biclustering could perform better. Indeed, some biclustering methods such as FABIA are frequently used in drug discovery (Verbist et al. 2015). An evaluation focussing on these kind of subproblems is still a possibility for future research.”

c) I agree with reviewer 1's original comment that an expression matrix clustered by rows and columns would still reveal a useful picture. I suggest the authors try to do this using two or three ways: one simple way is to cluster the rows and columns independently and order them. The other two ways can be based on SVD or one of the best performing decomposition based methods.

We included heatmaps of every dataset as Supplementary Figure 21. Each heatmap includes a hierarchical clustering and a principal component analysis. The modular nature of all datasets are evident based on both the hierarchical clustering and the decomposition, although for most datasets (especially human datasets) it is also clear that the data is much more complex than can be shown on a heatmap (with space constraints).

Please note that while Reviewer #3 doesn't have remarks to the authors, in his/her remarks to editors, he/she says his/her comments have been sufficiently addressed.

Reviewer #4 (Remarks to the Author):

I thank the authors for their clarification. I have to admit that I was wrong that no comparisons with standard clustering are performed in Hochreiter's FABIA paper. However, the authors' reply is not adequately addressing the point that I made. I was neither referring to small data sets in terms of the number of samples, nor was I referring to data sets in which the genes' fold changes are small. I was

referring to data sets in which only a small number of pathways/gene modules is differentially expressed, e.g. samples from the same cell lines, but in slightly different cell states or treated with different, but similar, drugs. This does not seem to be the case in the benchmark data sets considered in this study. Consider, for instance, the study of Verbist et al. (Drug Discovery Today 20, 2015) which also highlights that biclustering has become a standard part of the data analysis pipelines of pharma companies. I fear that the given paper will be misunderstood by the community in the way that biclustering is not a worthwhile tool at all, which is definitely not the case and has been demonstrated in many previous studies.

Thank you for the clarification. We agree that more subtle changes in expression, for example when manipulating cells with similar drugs, are not considered by the benchmark datasets which are used in our study. This has mainly to do with the way the gold standard is generated: because almost all gold standard modules would not be differentially expressed in such datasets, most would be undetectable, which would lead to a very large overestimation of the false negatives. As Reviewer #2 suggests, this is to be expected and is not an issue with the study.

Nonetheless, we agree that our results could lead to misinterpretations regarding the value of biclustering algorithms, and therefore added a disclaimer paragraph to the discussion discussing the point raised by the reviewer:

“Nonetheless, we acknowledge that our evaluation workflow still has some limitations for particular applications. Because we wanted to make sure that most of the modules present in our gold standard were also differentially expressed in the expression data, we used large expression compendia from very different biological conditions. However, this means that when expression differences are very subtle, other methods such as biclustering could perform better. Indeed, some biclustering methods such as FABIA are frequently used in drug discovery (Verbist et al. 2015). An evaluation focussing on these kind of subproblems is still a possibility for future research.”

REVIEWERS' COMMENTS:

Reviewer #2 (Remarks to the Author):

The authors have addressed all my comments. I have no further questions.

Reviewer #4 (Remarks to the Author):

I want to thank the authors for their revisions and their clarifications in the response letter! I am convinced that it is more than time to close this matter after quite a few rounds of revisions. Let me emphasize again that I very much appreciate the effort and diligence of this study. However, I have been concerned that the conclusions drawn from the results could be harmful in terms of biasing less knowledgeable users. I am glad about the disclaimer paragraph that has been added, but I would be even happier if the bold statement in the abstract "[...] the advantages of biclustering and network-inference based approaches are unclear when compared to clustering" would be re-formulated accordingly. I am convinced that this is possible by changing or adding only a few words. In any case, I rely on the authors' integrity and the editors' competence that this can be handled without the need to re-review the final revision.

We thank all reviewers for their constructive but critical comments in improving the manuscript.

Reviewer #2 (Remarks to the Author):

The authors have addressed all my comments. I have no further questions.

Thank you

Reviewer #4 (Remarks to the Author):

I want to thank the authors for their revisions and their clarifications in the response letter! I am convinced that it is more than time to close this matter after quite a few rounds of revisions. Let me emphasize again that I very much appreciate the effort and diligence of this study. However, I have been concerned that the conclusions drawn from the results could be harmful in terms of biasing less knowledgeable users. I am glad about the disclaimer paragraph that has been added, but I would be even happier if the bold statement in the abstract "[...] the advantages of biclustering and network-inference based approaches are unclear when compared to clustering" would be re-formulated accordingly. I am convinced that this is possible by changing or adding only a few words. In any case, I rely on the authors' integrity and the editors' competence that this can be handled without the need to re-review the final revision.

We thank the reviewer for his comments, and despite disagreements believe that the manuscript has been greatly improved thanks to his/her comments. We re-formulated the abstract as requested:

[...] Overall, decomposition methods outperform all other strategies, while we do not find a clear advantage of biclustering and network-inference based approaches on large gene expression datasets. [...]

We also want to mention that it is not our goal to scare away less knowledgeable users, and indeed recommend the top biclustering methods as methods when using modules to use expression data for unraveling function and disease.